# DIRECTIONAL DISTANCE FIELD FOR MODELING THE DIFFERENCE BETWEEN 3D POINT CLOUDS

## ABSTRACT

Quantifying the dissimilarity between two unstructured 3D point clouds is challenging yet essential, with existing metrics often relying on measuring the distance between corresponding points which can be either inefficient or ineffective. In this paper, we propose a novel distance metric called directional distance field (DDF), which computes the difference between the underlying 3D surfaces calibrated and induced by a set of reference points. By associating each reference point with two given point clouds through computing its directional distances to them, the difference in directional distances of an identical reference point characterizes the geometric difference between a typical local region of the two point clouds. Finally, DDF is obtained by averaging the directional distance differences of all reference points. We evaluate DDF on various optimization and unsupervised learning-based tasks, including shape reconstruction, rigid registration, scene flow estimation, and feature representation. Extensive experiments show that DDF achieves significantly higher accuracy under all tasks in a memory and computationally *efficient* manner, compared with existing metrics. As a generic metric, DDF can unleash the potential of optimization and learning-based frameworks for 3D point cloud processing and analysis. We include the source code in the supplementary material.

## 1 INTRODUCTION

3D point cloud data, which is a set of points defined by 3D coordinates to represent the geometric shape of an object or a scene, has been used in various fields, such as computer vision, 3D modeling, and robotics. Measuring the difference between 3D point clouds is critical in many tasks, e.g., reconstruction, rigid registration, etc. Different from 2D images, where pixel values are *structured* with regular 2D coordinates, allowing us to directly compute the difference between two images pixel-by-pixel, 3D point clouds are *unstructured*, i.e., there is no point-wise correspondence naturally available between two point

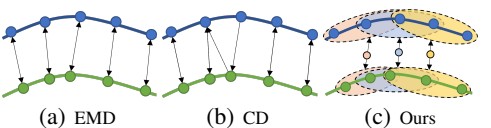

(a) EMD      (b) CD      (c) Ours

Figure 1: Visual illustration of different distance metrics for 3D point cloud data. 🔵 and 🟢 are two point clouds under evaluation, ▬ and ▬ are their underlying surfaces, and the lines with arrows indicate the correspondence. In contrast to existing metrics measuring the difference between corresponded points, our metric computes the difference between the local surfaces (the ovals) underlying 3D point clouds that are indirectly corresponded by reference points (the small circles) .

clouds, posing a great challenge. The two widely used distance metrics, namely Earth Mover's Distance (EMD) (Rubner et al., 2000) and Chamfer Distance (CD) (Barrow et al., 1977) illustrated in Figs. 1(a) and 1(b), first build point-wise correspondence between two point clouds and then compute the distance between corresponding points. However, EMD is both memory and time-consuming as it involves solving a linear programming problem for the optimal bijection. For each point in one point cloud, CD seeks its nearest point in the other point cloud to establish the correspondence, and it could easily reach a local minimum. Although some improved distance metrics (Wu et al., 2021; Nguyen et al., 2021; Deng et al., 2021; Urbach et al., 2020) have been proposed, they are still either inefficient or ineffective.

Existing distance metrics for 3D point cloud data generally concentrate on the point cloud itself, aligning points to measure the point-wise difference. However, these metrics overlook the fact that different point clouds obtained by different sampling could represent an identical 3D surface. In this paper, we propose an efficient yet effective distance metric named Directional Distance Field (DDF)

(DDF). *Unlike* previous metrics, DDF computes the difference between the underlying 3D surfaces of two point clouds, as depicted in Fig. 1(c). Specifically, we first sample a set of 3D points called reference points, which we use to induce and calibrate the local geometry of the surfaces underlying point clouds, i.e., computing the *directional distances* of each reference point to the two point clouds that approximately represent the underlying surfaces in implicit fields. Finally, we define DDF as the average of the directional distance differences of all reference points. We conduct extensive experiments on various tasks, including shape reconstruction, rigid registration, scene flow estimation, and feature representation, demonstrating its significant superiority over existing metrics.

In summary, the main contributions of this paper are:
1. an efficient, effective, and generic distance metric for 3D point cloud data;
2. state-of-the-art benchmarks of rigid registration and scene flow estimation.

## 2  RELATED WORK

**Distance Metrics for 3D Point Clouds.** Most of the existing distance metrics for 3D point clouds concentrate on the points in the point cloud. That is, they calculate the distance value based on the point-to-point distances from different point clouds, such as CD (Barrow et al., 1977) and EMD (Rubner et al., 2000). Specifically, EMD builds a global bi-directional mapping between the source and target point clouds. Then, the sum or mean of the distances between corresponding points is regarded as the distance between the point clouds. However, the computation of bijection is too expensive, especially when the number of points is large. Differently, CD builds a local mapping between point clouds by finding the nearest point in the other point cloud, making it more efficient than EMD. However, such local correspondence between point clouds may result in local minima or sub-optimal results. Hausdorff Distance (HD) (Huttenlocher et al., 1993) is modified from CD but focuses more on the outliers. Thus, it struggles to handle the details of point clouds and usually serves as an evaluation metric. Considering the distribution of point clouds, Wu et al. (2021) proposed density-aware CD (DCD) by introducing density information as weights into CD, achieving a higher tolerance to the outliers. Nguyen et al. (2021) proposed Sliced Wasserstein Distance (SWD) to measure the distance between point clouds, making it more efficient and effective than EMD and CD in the shape representation task. PointNetLK (Aoki et al., 2019) and FMR (Huang et al., 2020) convert point clouds to feature vectors through PointNet (Qi et al., 2017) and utilize the distance of features to measure the difference between point clouds. However, the global feature cannot represent the details of the point clouds, and such a kind of distance metric relies heavily on the encoder.

The above-mentioned distance metrics concentrate on points and ignore the geometric nature of a point cloud. In reality, the point clouds are sampled from the surface, and differently sampled point clouds could represent the same surface. Therefore, we should measure the difference between the underlying surfaces as the distance of the point clouds. ARL (Deng et al., 2021) randomly samples lines and calculates their intersections on the two point clouds' underlying surfaces approximately. It then calculates the difference between each line's intersections on the two point clouds to measure the dissimilarity of the two point clouds. However, the calculation of the intersection is time-consuming, and the randomness of the lines could also make the measurement unstable. To leverage the underlying surfaces of point clouds, DPDist (Urbach et al., 2020) trains a network to regress the point-to-plane distance between the two point clouds. However, the trained network's accuracy in regressing the point-to-plane distance would decrease if the distribution of point cloud changes, limiting its generalization. Other approaches, such as RMA-Net (Feng et al., 2021), employ the projection of point clouds onto 2D images and leverage image dissimilarity as a measure of the distance between point clouds. Nevertheless, this projection process is time-consuming and susceptible to occlusion, which imposes constraints on its practicality and application.

**Distance Metric-driven 3D Point Cloud Processing.** A distance metric is necessary for various 3D point cloud data processing and analysis tasks. One of the most common examples is the rigid registration of point clouds. The traditional registration methods, such as ICP (Besl & McKay, 1992) and its variants (Censi, 2008; Yang et al., 2013; Zhou et al., 2016), utilize the distance metrics between point clouds as the objective function to optimize. Some recent learning-based registration methods, such as DCP (Wang & Solomon, 2019), FMR (Huang et al., 2020), and RPM-Net (Yew & Lee, 2020), could become unsupervised with the distance metrics as the alignment item in their loss functions during training. Besides, recent learning-based methods for scene flow estimation, such as PointPWC-Net (Wu et al., 2020), NSFP (Li et al., 2021), and SCOOP (Lang et al., 2023),

adopt distance metrics as the alignment item in the loss functions to train the network without using ground-truth scene flow as supervision, and they have achieve remarkable accuracy. The distance metrics are also critical in some point cloud generation tasks, e.g., point cloud reconstruction (Yang et al., 2018; Groueix et al., 2018), upsampling (Yu et al., 2018a;b; Qian et al., 2021; 2020), completion (Yuan et al., 2018; Yu et al., 2021; Zhou et al., 2022; Xiang et al., 2021; Xie et al., 2020), etc., where the difference between the generated and ground-truth point clouds is calculated as the main loss to train the networks.

## 3 PROPOSED METHOD

### 3.1 PROBLEM STATEMENT AND OVERVIEW

Given any two unstructured 3D point clouds $\mathbf{P}_1 \in \mathbb{R}^{N_1 \times 3}$ and $\mathbf{P}_2 \in \mathbb{R}^{N_2 \times 3}$ with $N_1$ and $N_2$ points, respectively, we aim to construct a *differentiable* distance metric, which can quantify the difference between them effectively and efficiently to drive downstream tasks. As mentioned in Section 1, the problem is fundamentally challenging due to the lack of correspondence information between $\mathbf{P}_1$ and $\mathbf{P}_2$. Existing metrics generally focus on establishing the point-wise correspondence between $\mathbf{P}_1$ and $\mathbf{P}_2$ to compute the point-to-point difference, making them either ineffective or inefficient.

In contrast to existing metrics, we address this problem by measuring the difference between the underlying surfaces of $\mathbf{P}_1$ and $\mathbf{P}_2$. As shown in Fig. 2, with a set of reference points generated, we associate each reference point with $\mathbf{P}_1$ and $\mathbf{P}_2$ to implicitly model their local surface geometry through the proposed directional distance fields (DDFs). Then we calculate the difference between the local surface geometry reference point-by-reference point and average the differences of all reference points as the final distance between $\mathbf{P}_1$ and $\mathbf{P}_2$. Such a process is memory-saving, computationally efficient, and effective. In what follows, we will introduce the technical details of the proposed distance metric.

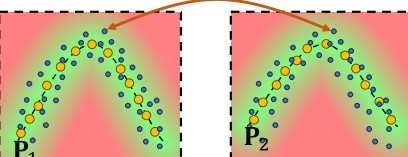

Figure 2: Overall illustration of the proposed distance metric in 2D condition. The orange points represent the point clouds under evaluation, the blue points represent the generated reference points, and the color inside the dashed rectangles represents the values of DDFs.

### 3.2 GENERATION OF REFERENCE POINTS

We generate a set of 3D points $\mathbf{Q} = \{\mathbf{q}_m \in \mathbb{R}^3\}_{m=1}^M$ named reference points, which will be used to *indirectly* establish the correspondence between $\mathbf{P}_1$ and $\mathbf{P}_2$ and induce the local geometry of the surfaces underlying $\mathbf{P}_1$ and $\mathbf{P}_2$ in an *implicit* manner. Technically, after selecting *either one* of $\mathbf{P}_1$ and $\mathbf{P}_2$[1] we add Gaussian noise to each point, where the standard deviation is $T$ times the distance to its nearest point in the point cloud to adapt the different densities at the location of each point. We repeat the noise addition process $R$ times randomly to generate $R$ reference points that are distributed *near* to the underlying surface. The whole generation process is shown in Fig. 3. See Table 8 for the ablative studies on this process.

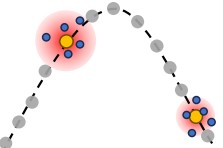

Figure 3: Illustration of the reference point generation. The radius of the red ball represents the standard deviation of the added Gaussian noise.

### 3.3 DIRECTIONAL DISTANCE FIELD

Let $\mathbf{P} \in \{\mathbf{P}_1, \mathbf{P}_2\}$, $\mathbf{q} \in \mathbf{Q}$, and $\Omega(\mathbf{q}, \mathbf{P}) := \{\mathbf{p}_k\}_{k=1}^K$ be the set of $\mathbf{q}$'s $K$-NN ($K$-Nearest Neighbor) points in $\mathbf{P}$. Note that we sort all points in $\Omega(\mathbf{q}, \mathbf{P})$ according to their distances to $\mathbf{q}$, i.e., $\|\mathbf{p}_1 - \mathbf{q}\|_2 \leq \|\mathbf{p}_2 - \mathbf{q}\|_2 \leq \dots \leq \|\mathbf{p}_K - \mathbf{q}\|_2$, where $\|\cdot\|_2$ is the $\ell_2$ norm of a vector. Denote by $\mathcal{S}$ the underlying surface of $\mathbf{P}$. We define an

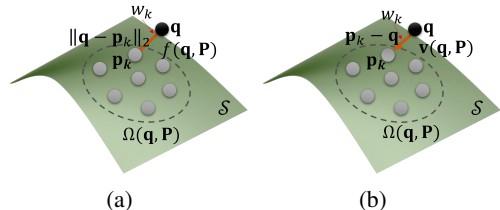

Figure 4: Illustration of the reference point-induced local surface geometry of a 3D point cloud. (a) $f(\mathbf{q}, \mathbf{P})$. (b) $\mathbf{v}(\mathbf{q}, \mathbf{P})$.

---

[1]For point cloud generation-based tasks, where one point cloud is used to supervise a network to generate the other point cloud, the one used as supervision will be selected to generate reference points because the generated one is messy in initial iterations, lacking sufficient geometric meaning.

implicit field, namely Directional Distance Field (DDF), to implicitly characterize $\mathcal{S}$. At reference point $\mathbf{q}$, DDF consists of a distance and a direction, denoted as $f(\mathbf{q}, \mathbf{P}) \in \mathbb{R}^1$ and $\mathbf{v}(\mathbf{q}, \mathbf{P}) \in \mathbb{R}^3$, respectively. Specifically, as depicted in Fig. 4(a), $f(\mathbf{q}, \mathbf{P})$ is defined as the weighted sum of the distances between $\mathbf{q}$ and each $\mathbf{p}_k \in \Omega(\mathbf{q}, \mathbf{P})$:

$$f(\mathbf{q}, \mathbf{P}) := \frac{\sum_{k=1}^K w(\mathbf{p}_k, \mathbf{q}) \cdot \|\mathbf{p}_k - \mathbf{q}\|_2}{\sum_{k=1}^K w(\mathbf{p}_k, \mathbf{q})}, \text{ where } w(\mathbf{p}_k, \mathbf{q}) = \frac{1}{\|\mathbf{p}_k - \mathbf{q}\|_2^2}. \tag{1}$$

Similarly, as shown in Fig. 4(b), $\mathbf{v}(\mathbf{q}, \mathbf{P})$ is defined as the weighted sum of the vectors from each $\mathbf{p}_k \in \Omega(\mathbf{q}, \mathbf{P})$ to $\mathbf{q}$:

$$\mathbf{v}(\mathbf{q}, \mathbf{P}) := \frac{\sum_{k=1}^K w(\mathbf{p}_k, \mathbf{q}) \cdot (\mathbf{p}_k - \mathbf{q})}{\sum_{k=1}^K w(\mathbf{p}_k, \mathbf{q})}. \tag{2}$$

Then, we concatenate $f(\mathbf{q}, \mathbf{P})$ and $\mathbf{v}(\mathbf{q}, \mathbf{P})$ to form a 4D vector $\mathbf{g}(\mathbf{q}, \mathbf{P}) = [f(\mathbf{q}, \mathbf{P}) \| \mathbf{v}(\mathbf{q}, \mathbf{P})] \in \mathbb{R}^4$ as the value of DDF at the location of $\mathbf{q}$. The DDF at all reference points could approximately characterizes $\mathcal{S}$, i.e., given $\mathbf{Q}$ associated with a DDF, the structure of $\mathcal{S}$ can be approximately inferred.

### 3.4 LOCAL SURFACE GEOMETRY-DRIVEN DISTANCE METRIC

We measure the dissimilarity between $\mathbf{P}_1$ and $\mathbf{P}_2$ by calculating the difference between their DDFs obtained through Eqs. (1) and (2). Specifically, for a typical reference point $\mathbf{q}_m \in \mathbf{Q}$, we have

$$d(\mathbf{q}_m, \mathbf{P}_1, \mathbf{P}_2) = \|\mathbf{g}(\mathbf{q}_m, \mathbf{P}_1) - \mathbf{g}(\mathbf{q}_m, \mathbf{P}_2)\|_1, \tag{3}$$

where $\| \cdot \|_1$ computes the $\ell_1$ norm of a vector. In practice, we notice that if the weights in $\mathbf{g}(\mathbf{q}_m, \mathbf{P}_1)$ and $\mathbf{g}(\mathbf{q}_m, \mathbf{P}_2)$ are derived independently using the inverse distance for two point clouds, for the point cloud to be optimized/generated, the directional distance will be a high-order non-linear function of its points, thereby complicating the optimization process. To avoid this, $\mathbf{g}(\mathbf{q}_m, \mathbf{P}_1)$ and $\mathbf{g}(\mathbf{q}_m, \mathbf{P}_2)$ should share *identical* weights $\{w\}_{k=1}^K$, where the weights are computed from $\mathbf{P}_1$ (resp. $\mathbf{P}_2$) and then shared with $\mathbf{P}_2$ (resp. $\mathbf{P}_1$) according to the task. Since the points in $\Omega(\mathbf{q}_m, \mathbf{P}_1)$ and $\Omega(\mathbf{q}_m, \mathbf{P}_2)$ are sorted in the same order, the $K$-NN points at the same rank in two point clouds share the same weights. The proposed distance metric for 3D point clouds is finally defined as the weighted sum of $d(\mathbf{q}_m, \mathbf{P}_1, \mathbf{P}_2)$ over all reference points:

$$\mathcal{D}_{\text{DDF}}(\mathbf{P}_1, \mathbf{P}_2) = \frac{1}{M} \sum_{\mathbf{q}_m \in \mathbf{Q}} s(\mathbf{q}_m, \mathbf{P}_1, \mathbf{P}_2) \cdot d(\mathbf{q}_m, \mathbf{P}_1, \mathbf{P}_2), \tag{4}$$

where $s(\mathbf{q}_m, \mathbf{P}_1, \mathbf{P}_2) = \text{Exp}(-\beta \cdot d(\mathbf{q}_m, \mathbf{P}_1, \mathbf{P}_2))$ is the confidence score of $d(\mathbf{q}_m, \mathbf{P}_1, \mathbf{P}_2)$ with $\beta \geq 0$ being a hyperparameter. Particularly, we introduce $s(\mathbf{q}_m)$ to cope with the case where $\mathbf{P}_1$ and $\mathbf{P}_2$ are partially overlapped.

***Remark***. According to Eqs. (3) and (4), it is obvious that the proposed DDF satisfies non-negativity, symmetry, and identity of indiscernibles, the essential properties possessed by a distance metric. Regarding triangle inequality, we refer readers to *Appendix* Sec. A for the proof. Second, our DDF showcases remarkable efficiency as it indirectly calibrates the point clouds through pre-defined reference points, a departure from the time-consuming correspondence matching approach used in previous methods. Also, our DDF is more effective by quantifying the disparity in underlying surfaces rather than focusing solely on point-wise differences.

## 4 EXPERIMENTS

To demonstrate the effectiveness and superiority of the proposed DDF, we applied it to a wide range of downstream tasks, including shape reconstruction, rigid point cloud registration, scene flow estimation, and feature representation. We made comparisons with the two well-known metrics i.e., EMD (Rubner et al., 2000) and CD (Barrow et al., 1977), and two recent metrics, i.e., DCD (Wu et al., 2021) and ARL (Deng et al., 2021)[2]. In all experiments, we set $R = 10$, $T = 3$, and $K = 5$. We set $\beta = 3$ in rigid registration due to the partially overlapping point clouds, and $\beta = 0$ in the other three tasks. We conducted all experiments on an NVIDIA RTX 3090 with Intel(R) Xeon(R) CPU.

---

[2]As ARL was primarily designed for rigid point cloud registration, ARL was evaluated only with the registration task in our experiments.

Table 1: Quantitative comparisons of reconstructed point clouds and surfaces. The best results are highlighted in **bold**. ↓ (resp. ↑) indicates the smaller (resp. the larger), the better.

| Shape | Loss | Point Cloud | | | Triangle Mesh | | |
|---|---|---|---|---|---|---|---|
| | | CD ($\times 10^{-2}$)↓ | HD ($\times 10^{-2}$)↓ | P2F ($\times 10^{-3}$)↓ | NC ↑ | F-0.5% ↑ | F-1% ↑ |
| Chair | EMD | 2.935 | 12.628 | 9.777 | 0.781 | 0.277 | 0.524 |
| | CD | 2.221 | 9.430 | 4.543 | 0.839 | 0.465 | 0.721 |
| | DCD | 3.020 | 10.755 | 4.164 | 0.892 | 0.674 | 0.868 |
| | Ours | 1.898 | 6.787 | 2.591 | 0.908 | 0.709 | 0.913 |
| Sofa | EMD | 2.534 | 7.355 | 7.433 | 0.879 | 0.459 | 0.717 |
| | CD | 1.972 | 5.323 | 3.392 | 0.920 | 0.668 | 0.887 |
| | DCD | 2.937 | 5.648 | 2.756 | 0.929 | 0.755 | 0.917 |
| | Ours | 1.770 | 4.191 | 2.040 | 0.940 | 0.806 | 0.949 |
| Table | EMD | 2.996 | 11.374 | 8.921 | 0.768 | 0.243 | 0.473 |
| | CD | 2.272 | 8.881 | 4.353 | 0.824 | 0.403 | 0.658 |
| | DCD | 2.250 | 8.031 | 4.643 | 0.868 | 0.577 | 0.789 |
| | Ours | 1.974 | 6.302 | 2.480 | 0.900 | 0.643 | 0.873 |
| **Average** | EMD | 2.821 | 10.452 | 8.720 | 0.809 | 0.326 | 0.571 |
| | CD | 2.155 | 7.878 | 4.096 | 0.861 | 0.521 | 0.755 |
| | DCD | 2.735 | 8.144 | 3.854 | 0.898 | 0.674 | 0.862 |
| | Ours | **1.880** | **5.760** | **2.370** | **0.916** | **0.719** | **0.911** |

## 4.1 3D SHAPE RECONSTRUCTION

We considered a learning-based point cloud shape reconstruction task. Technically, we followed Fold-ingNet (Yang et al., 2018) to construct a reconstruction network, where regular 2D grids distributed on a square area $[-\delta, \delta]^2$ are fed into MLPs to regress 3D point clouds. The network was trained by minimizing the distance between the reconstructed point cloud and the given point cloud in an overfitting manner, i.e., each shape has its individual network parameters. Additionally, based on the mechanism of the reconstruction network and differential geometry, we could obtain the normal vectors of the reconstructed point cloud through the backpropagation of the network. Finally, we used SPSR (Kazhdan & Hoppe, 2013) to recover the mesh from the resulting point cloud and its normal vectors. We refer the readers to *Appendix* Sec. B for more details.

**Implementation Details.** We utilized three categories of the ShapeNet dataset (Chang et al., 2015), namely chair, sofa, and table, each containing 200 randomly selected shapes. For each shape, we normalized it within a unit cube and sampled 4096 points from its surface uniformly using PDS (Bridson, 2007) to get the point cloud. As for the input regular 2D grids, we set $\delta = 0.3$ and the size to be $64 \times 64$. We used the ADAM optimizer to optimize the network for $10^4$ iterations with a learning rate of $10^{-3}$.

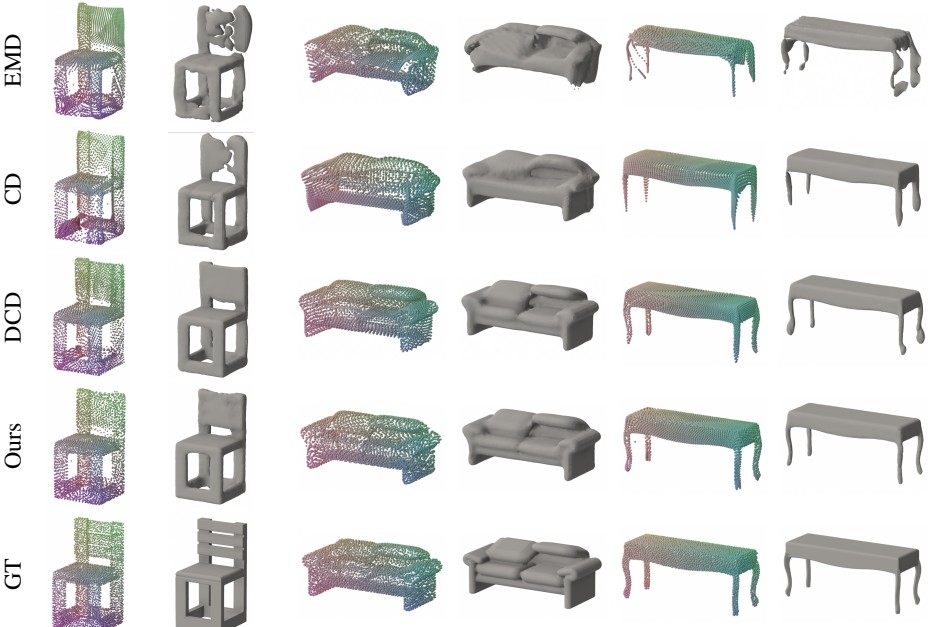

Figure 5: Visual comparisons of reconstructed shapes in the form of point clouds and surfaces under different distance metrics.

**Comparisons.** To quantitatively compare different distance metrics, we employed CD, HD, and the point-to-surface distance (P2F) to evaluate the accuracy of reconstructed point clouds. As for

Table 2: Quantitative comparisons of rigid registration on the Human dataset.

| | Method | Clean | | Noise | | Outlier | |
|---|---|---|---|---|---|---|---|
| | | RE (°) ↓ | TE (m) ↓ | RE (°) ↓ | TE (m) ↓ | RE (°) ↓ | TE (m) ↓ |
| Optimized | ICP | 1.276 | 0.068 | 6.790 | 0.236 | 37.515 | 0.590 |
| | FGR | 19.684 | 0.384 | 48.854 | 0.732 | 56.111 | 0.892 |
| | S-ICP | 1.587 | 0.056 | 3.754 | 0.168 | 16.757 | 0.261 |
| | EMD | 7.237 | 0.642 | 7.371 | 0.641 | 18.381 | 0.537 |
| | CD | 10.692 | 0.421 | 11.442 | 0.468 | 33.113 | 0.462 |
| | DCD | 40.973 | 0.933 | 41.457 | 0.942 | 41.671 | 0.937 |
| | ARL | 1.245 | 0.085 | 2.116 | 0.150 | 42.842 | 0.759 |
| | Ours | **1.040** | **0.040** | **1.745** | **0.106** | **3.991** | **0.080** |
| Unsupervised (RPM-Net) | EMD | 7.667 | 0.638 | 8.572 | 0.635 | 17.582 | 0.575 |
| | CD | 2.197 | 0.287 | 8.111 | 0.392 | 17.120 | 0.383 |
| | DCD | 8.735 | 0.623 | 5.862 | 0.184 | 20.344 | 0.217 |
| | ARL | 1.090 | 0.075 | 2.873 | 0.190 | 44.105 | 0.804 |
| | Ours | **0.793** | **0.053** | **2.253** | **0.128** | **4.593** | **0.124** |

the reconstructed triangle meshes, we used Normal Consistency (NC) and F-Score with thresholds of 0.5% and 1%, denoted as F-0.5% and F-1%, as the evaluation metrics. Table 1 and Fig. 5 show the numerical and visual results, respectively, where both quantitative accuracy and visual quality of reconstructed shapes by the network trained with our DDF are much better. Besides, due to the difficulty in establishing the optimal bijection between a relatively large

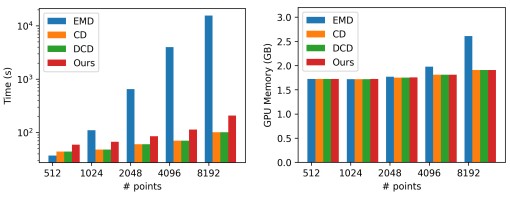

(a) Running time     (b) GPU Memory

Figure 6: Efficiency comparison under 3D shape reconstruction with various numbers of points.

number of points, i.e., 4096 points in our experiments, the network trained with EMD produces even worse shapes than that with CD and DCD. We included more results and analysis in *Appendix* Sec. B

**Efficiency Analysis.** We compared the running time in Fig. 6(a), where it can be seen that with the number of points increasing, the reconstruction driven by EMD requires much more time to optimize than that by CD and our DDF. Fig. 6(b) compares the GPU memory consumption, showing that these three distance metrics are comparable when the number of points is relatively small, but when dealing with a large number of points, EMD requires more GPU memory.

## 4.2 RIGID REGISTRATION

Given source and target point clouds, denoted as $\mathbf{P}_{\text{src}} \in \mathbb{R}^{N_{\text{src}} \times 3}$ and $\mathbf{P}_{\text{tgt}} \in \mathbb{R}^{N_{\text{tgt}} \times 3}$, respectively, rigid registration is to seek a spatial transformation $[\mathbf{R}, \mathbf{t}]$ to align $\mathbf{P}_{\text{src}}$ with $\mathbf{P}_{\text{tgt}}$, where $\mathbf{R} \in \text{SO}(3)$ is the rotation matrix and $\mathbf{t} \in \mathbb{R}^3$ is the translation vector. The optimization-based registration can be achieved by directly solving the following problem:

$$\{\hat{\mathbf{R}}, \hat{\mathbf{t}}\} = \underset{\mathbf{R}, \mathbf{t}}{\arg\min} \, \mathcal{D}\left(\mathcal{T}(\mathbf{P}_{\text{src}}, \mathbf{R}, \mathbf{t}), \mathbf{P}_{\text{tgt}}\right), \qquad (5)$$

where $\mathcal{D}(\cdot, \cdot)$ is the distance metric that can be EMD, CD, DCD, ARL, or our DDF, and $\mathcal{T}(\cdot, \cdot, \cdot)$ refers to applying the rigid transformation on a point cloud. We made comparisons with three well-known optimization-based rigid registration methods, i.e., ICP (Besl & McKay, 1992), FGR (Zhou et al., 2016), and S-ICP (Rusinkiewicz, 2019). Additionally, We also considered unsupervised learning-based rigid registration. Specifically, following (Deng et al., 2021), we modified RPM-Net (Yew & Lee, 2020), a supervised learning-based registration method, to be unsupervised by using a distance metric to drive the learning of the network. See *Appendix* Sec. C for more details.

**Implementation Details.** We used the Human dataset provided in (Deng et al., 2021), containing 5000 pairs of source and target point clouds for training and 500 pairs for testing. The source point clouds are 1024 points, while the target ones are 2048 points, i.e., they are partially overlapped. We used Open3D (Zhou et al., 2018) to implement ICP and FGR, and the official code released by the authors to implement S-ICP. For the optimization-based methods, we utilized Lie algebraic to represent the transformation and optimized it with the ADAM optimizer for 1000 iterations with a learning rate of 0.02. For the unsupervised learning-based methods, we kept the same training settings as (Deng et al., 2021).

**Comparison.** Following previous registration work (Choy et al., 2020), we adopted *Rotation Error (RE)* and *Translation Error (TE)* as the evaluation metrics. As shown in Table 2 and Fig. 7, our DDF outperforms the baseline methods in both optimization-based and unsupervised learning methods. ICP, FGR, S-ICP, as long as EMD, DCD and CD, can easily get local optimal solutions since they struggle to properly handle the outliers in the non-overlapping regions. ARL (Deng et al., 2021)

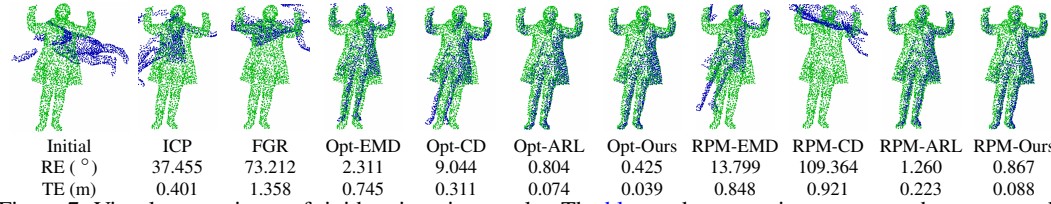

| | Initial | ICP | FGR | Opt-EMD | Opt-CD | Opt-ARL | Opt-Ours | RPM-EMD | RPM-CD | RPM-ARL | RPM-Ours |
|---|---|---|---|---|---|---|---|---|---|---|---|
| RE ( $^{\circ}$ ) | 37.455 | 73.212 | 2.311 | 9.044 | 0.804 | 0.425 | 13.799 | 109.364 | 1.260 | 0.867 | |
| TE (m) | 0.401 | 1.358 | 0.745 | 0.311 | 0.074 | 0.039 | 0.848 | 0.921 | 0.223 | 0.088 | |

Figure 7: Visual comparisons of rigid registration results. The blue and green points represent the source and target point clouds, respectively. Opt-X represents the optimization-based method in Eq. (5) with a chosen distance metric. 🔍 Zoom in to see details.

and our DDF do not have such a disadvantage, but the randomly sampled lines in ARL decrease the registration accuracy.

**Efficiency Analysis.** Table 3 lists the running time and GPU memory costs of the optimization-based registration driven by different metrics[3]. Since the number of points is relatively small, EMD has comparable running time and GPU memory consumption to CD, DCD, and our DDF. In contrast, ARL (Deng et al., 2021) is much less efficient because of the calculation of the intersections.

Table 3: Running time (s) and GPU memory (GB) costs of different distance metrics under the optimization-based rigid registration task.

| | Time | GPU Memory |
|---|---|---|
| EMD | 11 | 1.680 |
| CD | 11 | 1.809 |
| DCD | 12 | 1.809 |
| ARL | 281 | 7.202 |
| Ours | 13 | 1.813 |

### 4.3 SCENE FLOW ESTIMATION

This task aims to predict point-wise offsets $\mathbf{F} \in \mathbb{R}^{N_{\mathrm{src}} \times 3}$, which can align source point cloud $\mathbf{P}_{\mathrm{src}}$ to target point cloud $\mathbf{P}_{\mathrm{tgt}}$. The optimization-based methods directly solve

$$\hat{\mathbf{F}} = \arg \min_{\mathbf{F}} \mathcal{D}(\mathbf{P}_{\mathrm{src}} + \mathbf{F}, \mathbf{P}_{\mathrm{tgt}}) + \alpha \mathcal{R}_{\mathrm{smooth}}(\mathbf{F}), \tag{6}$$

where $\mathcal{R}_{\mathrm{smooth}}(\cdot)$ is the spatial smooth regularization term and the hyperparameter $\alpha > 0$ balances the two iterms. Besides, we also evaluated the proposed DDF by incorporating it into unsupervised learning-based frameworks, i.e., replacing the distance metrics of existing unsupervised learning-based frameworks with our DDF to train the network, We adopted two state-of-the-art unsupervised learning-based scene flow estimation methods named NSFP (Li et al., 2021) and SCOOP (Lang et al., 2023), both of which utilize CD to drive the learning of the network.

**Implementation Details.** We used the Flyingthings3D dataset (Mayer et al., 2016) preprocessed by (Gu et al., 2019), where point clouds each with $N_{\mathrm{src}} = N_{\mathrm{tgt}} = 8192$ points were sampled randomly and *non-uniformly* distributed. For the optimization-based methods, we used $\ell_2$-smooth regularization, set $\alpha = 50$, and optimized the scene flow directly with the ADAM optimizer for 500 iterations with a learning rate of 0.01. For the unsupervised learning-based methods, we adopted the same training settings as their original papers.

Table 4: Quantitative comparisons of scene flow estimation on the Flyingthings3D dataset .

| | Method | EPE3D(m)↓ | Acc-0.05 ↑ | Acc-0.1↑ | Outliers ↓ |
|---|---|---|---|---|---|
| Optimized | EMD | 0.3681 | 0.1894 | 0.4226 | 0.7838 |
| | CD | 0.1557 | 0.3489 | 0.6581 | 0.6799 |
| | DCD | 0.7045 | 0.0309 | 0.0980 | 0.9965 |
| | Ours | **0.0843** | **0.5899** | **0.8722** | **0.4707** |
| Unsupervised | NSFP | 0.0899 | 0.6095 | 0.8496 | 0.4472 |
| | NSFP + Ours | **0.0662** | **0.7346** | **0.9107** | **0.3426** |
| | SCOOP | 0.0839 | 0.5698 | 0.8516 | 0.4834 |
| | SCOOP + Ours | **0.0742** | **0.6134** | **0.8858** | **0.4497** |

**Comparison.** Following (Li et al., 2021; Lang et al., 2023), we employed *End Point Error (EPE)*, *Flow Estimation Accuracy (Acc)* with thresholds 0.05 and 0.1 (denoted as *Acc-0.05* and *Acc-0.1*), and *Outliers* as the evaluation metrics. From the results shown in Table 4 and Fig. 8, it can be seen that our DDF drives much more accurate scene flows than EMD, CD, and DCD under the optimization-based framework, and our DDF further boosts the accuracy of SOTA unsupervised learning-based methods to a significant extent, demonstrating its superiority and the importance of the distance metric in 3D point cloud modeling.

---

[3]Note that for the unsupervised learning-based methods, different distance metrics are only used to train the network, and the inference time of a trained network with different metrics is equal.

| | GT | Opt-EMD | Opt-CD | Opt-Ours | NSFP | NSFP + Ours | SCOOP | SCOOP + Ours |
|---|---|---|---|---|---|---|---|---|
| EPE3D (m) | 0.247 | 0.079 | 0.067 | 0.086 | 0.050 | 0.071 | 0.047 |
| Acc-0.05 | 0.437 | 0.627 | 0.875 | 0.527 | 0.859 | 0.648 | 0.892 |
| Acc-0.1 | 0.672 | 0.963 | 0.973 | 0.974 | 0.991 | 0.989 | 0.995 |
| Outliers | 0.361 | 0.069 | 0.044 | 0.057 | 0.010 | 0.040 | 0.019 |

Figure 8: Visual comparisons of scene flow estimation. The blue and green points represent the source and target point clouds, respectively, and the red points are the warped source point cloud with estimated scene flows. 🔍 Zoom in to see details.

**Efficiency Analysis.** We also compared the running time and GPU memory cost of different metrics in the optimization-based framework. As shown in Table 5, EMD consumes much more time and GPU memory than CD and our DDF.

Table 5: Running time (s) and GPU memory (GB) costs of different distance metrics under the optimization-based scene flow estimation task.

| | Time | GPU Memory |
|---|---|---|
| EMD | 1011 | 2.204 |
| CD | 7 | 1.704 |
| DCD | 8 | 1.704 |
| Ours | 14 | 1.725 |

## 4.4 FEATURE REPRESENTATION

In this experiment, we trained an auto-encoder with different distance metrics used as the reconstruction loss to evaluate their abilities. Technically, an input point cloud is encoded into a global feature through the encoder, which is further decoded to reconstruct the input point cloud through a decoder. After training, we used the encoder to represent point clouds as features for classification by an SVM classifier.

**Implementation Details.** We built an auto-encoder with MLPs and used the ShapeNet (Chang et al., 2015) and ModelNet40 (Wu et al., 2015) datasets for training and testing, respectively. We trained the network for 300 epochs using the ADAM optimizer with a learning rate of $10^{-3}$. We refer the readers to *Appendix* Sec. E for more details.

**Comparison.** As listed in Table 6, the higher classification accuracy by our DDF demonstrates the auto-encoder driven by our DDF can learn more discriminative features. Besides, we also used T-SNE (Van der Maaten & Hinton, 2008) to visualize the 2D embeddings of the global features of 1000 shapes from 10 categories in Fig. 9 to show the advantages of our method more intuitively.

Table 6: Classification accuracy by SVM on ModelNet40.

| Loss Function | EMD | CD | Ours |
|---|---|---|---|
| Accuracy (%) | 78.12 | 78.89 | 81.28 |

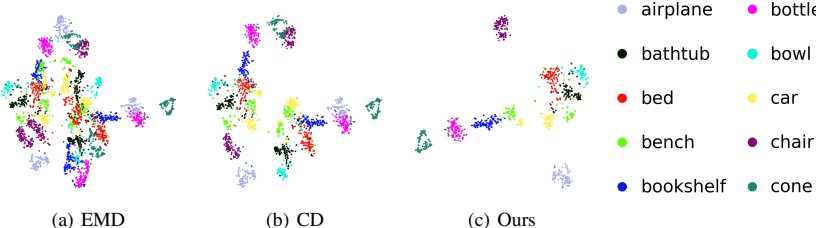

| (a) EMD | (b) CD | (c) Ours |
|---|---|---|

airplane · bottle
· bathtub · bowl
· bed · car
· bench · chair
· bookshelf · cone

Figure 9: The T-SNE clustering visualization of the features obtained from the auto-encoder trained with different distance metrics.

## 4.5 ABLATION STUDY

We carried out comprehensive ablation studies on shape reconstruction and rigid registration to verify the rationality of the design in our DDF.

**Necessity of shared weights.** When calculating the DDFs of the two point clouds, their weights in Eqs. (1) and (2) are shared, which can benefit the modeling accuracy. To verify this, we conducted an ablative study, i.e., their weights are independently calculated. The numerical results shown in Table 7, demonstrate the advantage of using shared weights for the two point clouds.

Table 7: Quantitative analysis of the effect of DDF with independent and shared weights on shape reconstruction.

| | CD ($\times 10^{-2}$) ↓ | HD ($\times 10^{-2}$) ↓ | P2F ($\times 10^{-3}$) ↓ | NC ↑ | F-0.5% ↑ | F-1% ↑ |
|---|---|---|---|---|---|---|
| Independent Weights | 1.964 | 5.939 | 2.388 | 0.898 | 0.631 | 0.848 |
| Shared Weights | 1.880 | 5.760 | 2.370 | 0.916 | 0.719 | 0.911 |

**Necessity of** $f(\cdot, \cdot)$ **and** $\mathbf{v}(\cdot, \cdot)$. We removed $f$ or $\mathbf{v}$ from the directional distance $\mathbf{g}$ when modeling the local surface geometry in Section 3.3. The accuracy of reconstructed 3D mesh shapes is listed in Table 8, where it can be seen that removing either one of $f$ and $\mathbf{v}$ would decrease the reconstruction accuracy, especially $\mathbf{v}$, because using $f$ or $\mathbf{v}$ alone could hardly characterize the underlying surfaces of point clouds.

Table 8: Reconstruction accuracy under different choices of directional distance.

| Geometry | NC ↑ | F-0.5% ↑ | F-1% ↑ |
|---|---|---|---|
| $f$ | 0.791 | 0.611 | 0.776 |
| $\mathbf{v}$ | 0.907 | 0.675 | 0.888 |
| $\mathbf{g}$ | 0.916 | 0.719 | 0.911 |

**Size of** $\Omega(\mathbf{q}, \mathbf{P})$. We changed the size of $\Omega(\mathbf{q}, \mathbf{P})$ by varying the value of $K$ used in Eqs. (1) and (2). As shown in Table 9, a small or large value of $K$ would decrease the performance of DDF because a small $K$ only covers a tiny region on the point cloud, resulting in $\mathbf{g}(\mathbf{q}, \mathbf{P})$ being unable to represent the local surface geometry induced by $\mathbf{q}$, while a large $K$ includes too many points, making $\mathbf{g}(\mathbf{q}, \mathbf{P})$ would become smooth and ignore the details of the local surface geometry induced by $\mathbf{q}$.

Table 9: The effect of $K$ on reconstruction accuracy.

| $K$ | NC ↑ | F-0.5% ↑ | F-1% ↑ | Time (s) ↓ |
|---|---|---|---|---|
| 1 | 0.913 | 0.687 | 0.888 | 119 |
| 3 | 0.916 | 0.717 | 0.908 | 124 |
| 5 | 0.916 | 0.719 | 0.911 | 126 |
| 10 | 0.913 | 0.712 | 0.907 | 132 |

**Reference Points.** We studied how the number of reference points affects DDF by varying the value of $R$. As shown in Table 10, too few reference points cannot sufficiently capture the local surface geometry difference, while too many reference points are not necessary for improving performance but compromise efficiency. When $R = 1$, our DDF has the same computational complexity as CD. In this case, our DDF still outperforms CD, thereby demonstrating that the superior performance of our DDF is primarily attributed to its well-designed structure rather than an increase in computational complexity. Moreover, we also studied how the distribution of reference points affects DDF by varying the value of $T$, concluding that the reconstruction accuracy decreases if the reference points are either too close to or too far from the underlying surface.

Table 10: Reconstruction accuracy under different settings of the reference points.

| $R$ | $T$ | NC ↑ | F-0.5% ↑ | F-1% ↑ | Time (s) ↓ |
|---|---|---|---|---|---|
| 1 | 3 | 0.899 | 0.610 | 0.831 | 110 |
| 5 | 3 | 0.912 | 0.686 | 0.886 | 112 |
| 20 | 3 | 0.913 | 0.693 | 0.893 | 149 |
| 10 | 1 | 0.916 | 0.722 | 0.909 | 126 |
| 10 | 5 | 0.911 | 0.692 | 0.896 | 126 |
| 10 | 10 | 0.891 | 0.610 | 0.836 | 126 |
| 10 | 3 | 0.916 | 0.719 | 0.911 | 126 |

**Effectiveness of** $s(\mathbf{q})$ **in Rigid Registration.** We set various values of $\beta$ in Eq. (4) in the rigid registration task. As shown in Table 11, when $\beta = 0$, the registration accuracy decreases significantly because the source and target point clouds are partially overlapped, and those reference points, which correspond to the non-overlapping regions and negate the optimization process, are still equally taken into account. On the contrary, if the value of $\beta$ is large, the registration accuracy only drops slightly. The reason is that the weights of the reference points with similar local surface geometry differences would be quite different, and the details of the point clouds may be ignored, making the registration accuracy slightly affected.

Table 11: Effect of the value of $\beta$ on rigid registration accuracy.

| $\beta$ | RE (°) | TE (m) |
|---|---|---|
| 0 | 5.637 | 0.286 |
| 1 | 1.598 | 0.080 |
| 3 | 1.040 | 0.040 |
| 5 | 1.290 | 0.037 |
| 10 | 1.769 | 0.046 |

## 5 CONCLUSION

We have introduced DDF, a novel, robust, and generic distance metric for 3D point clouds. DDF measures the difference between the local surfaces underlying 3D point clouds under evaluation, which is significantly different from existing metrics. Our extensive experiments have demonstrated the significant superiority of DDF in terms of both efficiency and effectiveness for various tasks, including shape reconstruction, rigid registration, scene flow estimation, and feature representation. Besides, comprehensive ablation studies have validated its rationality. DDF can further unleash the potential of optimization and learning-based frameworks to advance the field of 3D point cloud processing and analysis.

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

# Appendix

## A    PROOF OF TRIANGLE INEQUALITY

Given three sets of point clouds denoted as $\mathbf{P}_1$, $\mathbf{P}_2$, and $\mathbf{P}_3$, along with the generated reference point set $\mathbf{Q}$, our objective is to demonstrate the validity of the following inequality:

$$\mathcal{D}_{\text{DDF}}(\mathbf{P}_1, \mathbf{P}_2) + \mathcal{D}_{\text{DDF}}(\mathbf{P}_2, \mathbf{P}_3) \geq \mathcal{D}_{\text{DDF}}(\mathbf{P}_1, \mathbf{P}_3). \tag{7}$$

For ease of description, we denote the differences introduced by a reference point $\mathbf{q}_m \in \mathbf{Q}$, $d(\mathbf{q}_m, \mathbf{P}_1, \mathbf{P}_3)$, $d(\mathbf{q}_m, \mathbf{P}_1, \mathbf{P}_2)$ and $d(\mathbf{q}_m, \mathbf{P}_2, \mathbf{P}_3)$, as $d_{13}$, $d_{12}$, and $d_{23}$, respectively. According to the triangle inequality, these three item satisfies

$$d_{12} + d_{23} \geq d_{13}. \tag{8}$$

We denote $z(x) = \text{Exp}(-\beta x) \cdot x$, and we only need to prove the following inequality

$$z(d_{12}) + z(d_{23}) \geq z(d_{13}).$$

As outlined in Section 3.2, the reference points are generated in proximity to the underlying surfaces of the point clouds. Consequently, the differences introduced by the reference point $\mathbf{q}_m$ are sufficiently small, such that they are less than $\frac{1}{\beta}$, i.e., $d_{13}, d_{12}, d_{23} < \frac{1}{\beta}$.

On the other hand, we notice that the function $z(x)$ has the following two properties:

- $z(x)$ is monotonically increasing in the period $(0, \frac{1}{\beta})$,

- for any non-negative constant $a \leq 1$, it satisfies $z(ax) \geq a \cdot z(x)$ in the period $(0, \frac{1}{\beta})$.

If at least one of $d_{12}$ and $d_{23}$ is greater than $d_{13}$, the triangle inequality obviously holds. If both $d_{12}$ and $d_{23}$ are not greater than $d_{13}$, they could be represented as $d_{12} = a_1 d_{13}$ and $d_{23} = a_2 d_{13}$, where $0 < a_1, a_2 \leq 1$ and $a_1 + a_2 \geq 1$. According to the second property of $z(x)$, there are

$$z(d_{12}) = z(a_1 d_{13}) \geq a_1 \cdot z(d_{13}),$$
$$z(d_{23}) = z(a_2 d_{13}) \geq a_2 \cdot z(d_{13}).$$

By adding these two inequalities, we have

$$z(d_{12}) + z(d_{23}) \geq (a_1 + a_2) \cdot z(d_{13})$$
$$\geq z(d_{13}).$$

## B  SHAPE RECONSTRUCTION

**Network Architecture.** Fig. 10 shows the architecture of the neural network $\mathcal{F}(\cdot) : \mathbb{R}^2 \to \mathbb{R}^3$ used for 3D shape reconstruction, which maps a 2D grid $\mathbf{u} := [u_1, \ u_2]^\mathsf{T}$ to a 3D coordinate $\mathbf{x} := [x, \ y, \ z]^\mathsf{T}$, i.e., $\mathbf{x} = \mathcal{F}(\mathbf{u})$.

**Normal Estimation.**

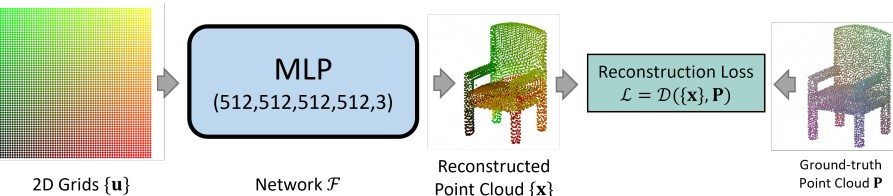

Figure 10: The network architecture used for 3D shape reconstruction.

According to the differential geometric property of $\mathcal{F}$, we could obtain the normal vector of a typical 3D point $\mathbf{x}$ corresponding to $\mathbf{u}$ by

$$\mathbf{n}(\mathbf{x}) = \frac{\frac{\partial \mathcal{F}(\mathbf{u})}{\partial u_1} \times \frac{\partial \mathcal{F}(\mathbf{u})}{\partial u_2}}{\left\| \frac{\partial \mathcal{F}(\mathbf{u})}{\partial u_1} \times \frac{\partial \mathcal{F}(\mathbf{u})}{\partial u_2} \right\|_2}, \tag{9}$$

where $\times$ is the cross product of two vectors, and the partial derivatives can be calculated through the backpropagation of the network. The resulting normal vectors as well as the 3D coordinates are used for mesh reconstruction via SPSR (Kazhdan & Hoppe, 2013).

**Evaluation Metric.** Let $\mathcal{S}_{\mathrm{rec}}$ and $\mathcal{S}_{\mathrm{GT}}$ denote the reconstructed and ground-truth 3D meshes, respectively, on which we randomly sample $N_{\mathrm{eval}} = 10^5$ points, denoted as $\mathbf{P}_{\mathrm{rec}}$ and $\mathbf{P}_{\mathrm{GT}}$. For each point on $\mathbf{P}_{\mathrm{rec}}$ and $\mathbf{P}_{\mathrm{GT}}$, the normal of the triangle face where it is sampled is considered to be its normal vector, and the normal sets of $\mathbf{P}_{\mathrm{rec}}$ and $\mathbf{P}_{\mathrm{GT}}$ are denoted as $\mathbf{N}_{\mathrm{rec}}$ and $\mathbf{N}_{\mathrm{GT}}$, respectively.

Let $\mathtt{NN\_Norm}(\mathbf{x}, \mathbf{P})$ be the operator that returns the normal vector of the point $\mathbf{x}$'s nearest point in the point cloud $\mathbf{P}$. The NC is defined as

$$\mathrm{NC}(\mathcal{S}_{\mathrm{rec}}, \mathcal{S}_{\mathrm{GT}}) = \frac{1}{N_{\mathrm{eval}}} \sum_{\mathbf{x} \in \mathbf{P}_{\mathrm{rec}}} |\mathbf{N}_{\mathrm{rec}}(\mathbf{x}) \cdot \mathtt{NN\_Normal}(\mathbf{x}, \mathbf{P}_{\mathrm{GT}})|$$
$$+ \frac{1}{N_{\mathrm{eval}}} \sum_{\mathbf{x} \in \mathbf{P}_{\mathrm{GT}}} |\mathbf{N}_{\mathrm{GT}}(\mathbf{x}) \cdot \mathtt{NN\_Normal}(\mathbf{x}, \mathbf{P}_{\mathrm{rec}})|. \tag{10}$$

The F-Score is defined as the harmonic mean between the precision and the recall of points that lie within a certain distance threshold $\epsilon$ between $\mathcal{S}_{\mathrm{rec}}$ and $\mathcal{S}_{\mathrm{GT}}$,

$$\mathtt{F\text{-}Score}(\mathcal{S}_{\mathrm{rec}}, \mathcal{S}_{\mathrm{GT}}, \epsilon) = \frac{2 \cdot \mathtt{Recall} \cdot \mathtt{Precision}}{\mathtt{Recall} + \mathtt{Precision}}, \tag{11}$$

where

$$\mathtt{Recall}(\mathcal{S}_{\mathrm{rec}}, \mathcal{S}_{\mathrm{GT}}, \epsilon) = \left| \left\{ \mathbf{x}_1 \in \mathbf{P}_{\mathrm{rec}}, \text{ s.t. } \min_{\mathbf{x}_2 \in \mathbf{P}_{\mathrm{GT}}} \|\mathbf{x}_1 - \mathbf{x}_2\|_2 < \epsilon \right\} \right|,$$

$$\mathtt{Precision}(\mathcal{S}_{\mathrm{rec}}, \mathcal{S}_{\mathrm{GT}}, \epsilon) = \left| \left\{ \mathbf{x}_2 \in \mathbf{P}_{\mathrm{GT}}, \text{ s.t. } \min_{\mathbf{x}_1 \in \mathbf{P}_{\mathrm{rec}}} \|\mathbf{x}_1 - \mathbf{x}_2\|_2 < \epsilon \right\} \right|.$$

**More visual results.** Figure 11 provides more visual results illustrating the reconstructed point clouds and meshes under different distance metrics. Obviously, our method has superior performance. Notably, the meshes are reconstructed utilizing SPSR Kazhdan & Hoppe (2013), where the accuracy of normals is crucial. Consequently, while certain methods may produce reasonably accurate point clouds, the meshes they generate may manifest considerable errors due to inaccuracies in the estimation of normals.

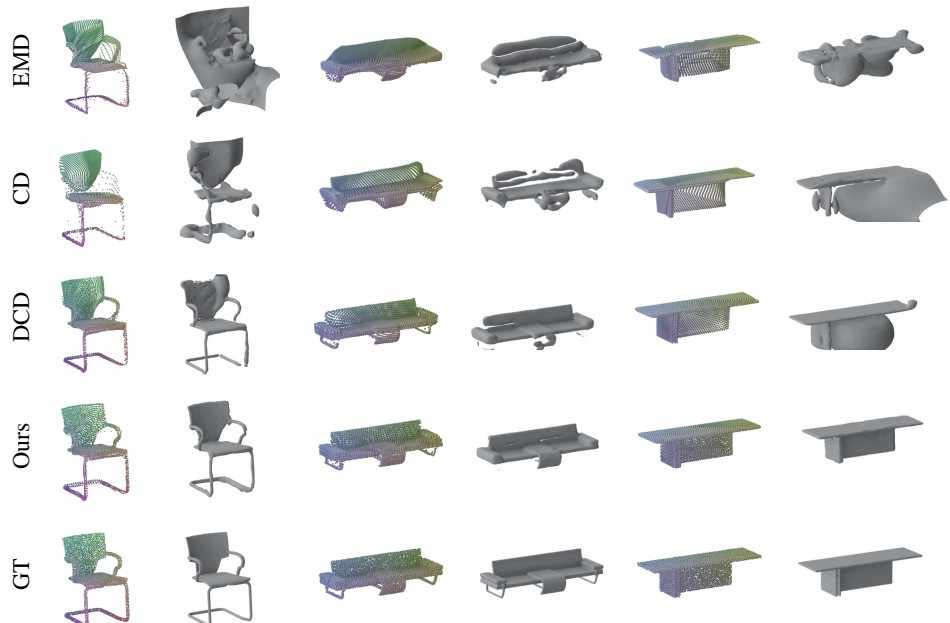

Figure 11: More visual comparisons of reconstructed shapes in the form of point clouds and surfaces under different distance metrics.

**Error & Accuracy Distribution.** The distributions of the reconstruction error and accuracy are illustrated in Fig. 12. Notably, it is evident that the 3D reconstruction supervised under our distance metric exhibits superior performance for both reconstructed point clouds and meshes.

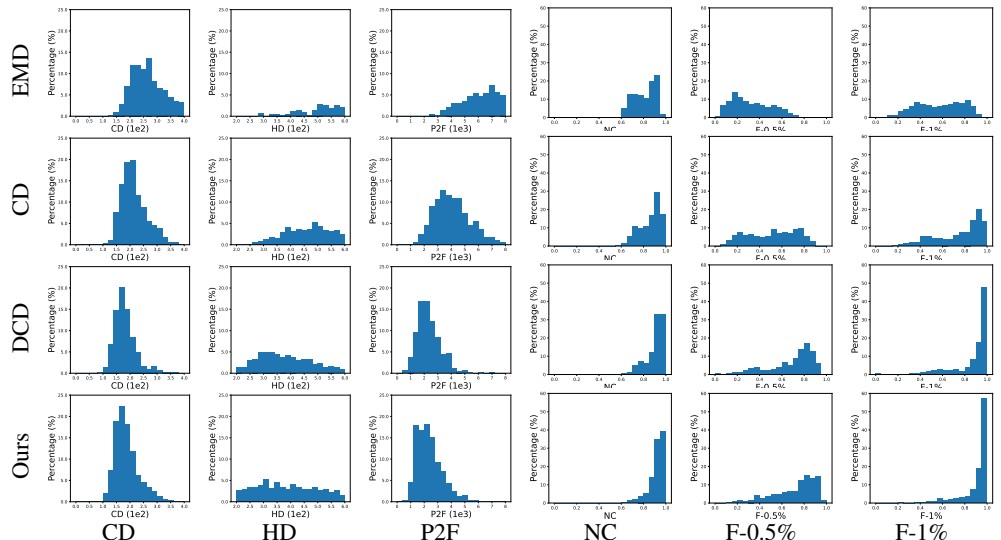

Figure 12: Histograms of different evaluation metrics (CD, HD, P2F, NC, F-0.5%, and F-1%) on the 3D shape reconstruction experiment under different distance metrics.

## C  RIGID REGISTRATION

**Evaluation Metric.** Let $[\hat{\mathbf{R}}, \hat{\mathbf{t}}]$ and $[\mathbf{R}_{\mathrm{GT}}, \mathbf{t}_{\mathrm{Gt}}]$ be the estimated and ground-truth transformation, respectively. The two evaluation metrics named Rotation Error (RE) and Translation Error (TE) are defined as

$$\mathrm{RE}(\hat{\mathbf{R}}, \mathbf{R}_{\mathrm{GT}}) = \angle(\mathbf{R}_{\mathrm{GT}}^{-1}\hat{\mathbf{R}}), \quad \mathrm{TE}(\hat{\mathbf{t}}, \mathbf{t}_{\mathrm{GT}}) = \|\hat{\mathbf{t}} - \mathbf{t}_{\mathrm{GT}}\|_2, \tag{12}$$

where $\angle(\mathbf{A}) = \arccos(\frac{\mathrm{trace}(\mathbf{A})-1}{2})$ returns the angle of rotation matrix $\mathbf{A}$ in degrees.

**Unsupervised Learning-based Method.**

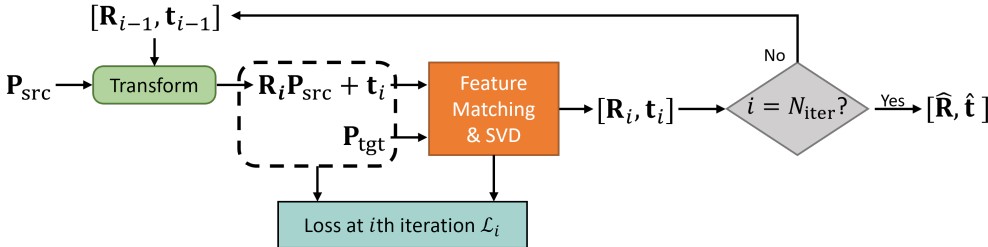

Figure 13: The overall pipeline of the unsupervised learning-based rigid registration built upon RPM-Net (Yew & Lee, 2020). The loss $\mathcal{L}_i$ is defined in Eq. equation 13. $N_{\mathrm{iter}}$ is the total number of iterations, and $[\mathbf{R}_i, \mathbf{t}_i]$ is the resulting transformation at the $i$-th iteration. See the original paper for detailed structures.

Based on RPM-Net (Yew & Lee, 2020), we construct an unsupervised learning-based rigid registration framework by modifying its loss, as shown in Fig. 13. The loss at $i$-th iteration $\mathcal{L}_i$ is defined as

$$\mathcal{L}_i = \lambda_1 \mathcal{D}(\mathbf{R}_i \mathbf{P}_{\mathrm{src}} + \mathbf{t}_i, \mathbf{P}_{\mathrm{tgt}}) + \lambda_2 \mathcal{R}_{\mathrm{inlier}}(\mathbf{P}_{\mathrm{src}}, \mathbf{P}_{\mathrm{tgt}}), \tag{13}$$

where $\mathcal{R}_{\mathrm{inlier}}$ is the inlier regularization in RPM-Net (see Eq. (11) of the original paper of RPM-Net for more details), and the hyperparameters, $\lambda_1$ and $\lambda_2$, are set 10 and 0.01, respectively. Considering all iterations, the total loss is

$$\mathcal{L}_{\mathrm{total}} = \sum_{i=1}^{N_{\mathrm{iter}}} \left(\frac{1}{2}\right)^{(N_{\mathrm{iter}}-i)} \mathcal{L}_i, \tag{14}$$

assigning later iterations with higher weights. We set $N_{\mathrm{iter}} = 2$ and $N_{\mathrm{iter}} = 5$ during training and inference, respectively.

**Error Distribution.** The distributions of the registration error are shown in Fig. 14, where it can be seen that under both optimization and unsupervised learning pipelines, the registration errors by our methods are concentrated in smaller ranges, compared with other methods.

## D  SCENE FLOW ESTIMATION

**Spatial Smooth Regularization.** Given the source point cloud $\mathbf{P}_{\mathrm{src}} \in \mathbb{R}^{N_{\mathrm{src}} \times 3}$ and its estimated scene flow $\mathbf{F} \in \mathbb{R}^{N_{\mathrm{src}} \times 3}$, we define the spatial smooth regularization term $\mathcal{R}_{\mathrm{smooth}}(\mathbf{F})$ as

$$\mathcal{R}_{\mathrm{smooth}}(\mathbf{F}) = \frac{1}{3N_{\mathrm{src}}K_{\mathrm{s}}} \sum_{\mathbf{x} \in \mathbf{P}_{\mathrm{src}}} \sum_{\mathbf{x}' \in \mathcal{N}(\mathbf{x})} \|\mathbf{F}(\mathbf{x}) - \mathbf{F}(\mathbf{x}')\|_2^2, \tag{15}$$

where $\mathcal{N}(\mathbf{x})$ is the operator returning $\mathbf{x}$'s $K_{\mathrm{s}}$-NN points in the $\mathbf{P}_{\mathrm{src}}$ with $K_{\mathrm{s}} = 30$ in our experiments.

## E  FEATURE REPRESENTATION

**Network Architecture.** Fig. 15 shows the network architecture of the auto-decoder for feature representation, where the encode embeds an input point cloud $\mathbf{P}$ as a global feature $\mathbf{z}$, and the resulting global feature is then fed into the decoder to get the reconstructed point cloud $\hat{\mathbf{P}}$. The difference between $\mathbf{P}$ and $\hat{\mathbf{P}}$ is employed as the loss to train the auto-decoder:

$$\mathcal{L}_{\mathrm{rec}} = \mathcal{D}(\hat{\mathbf{P}}, \mathbf{P}), \tag{16}$$

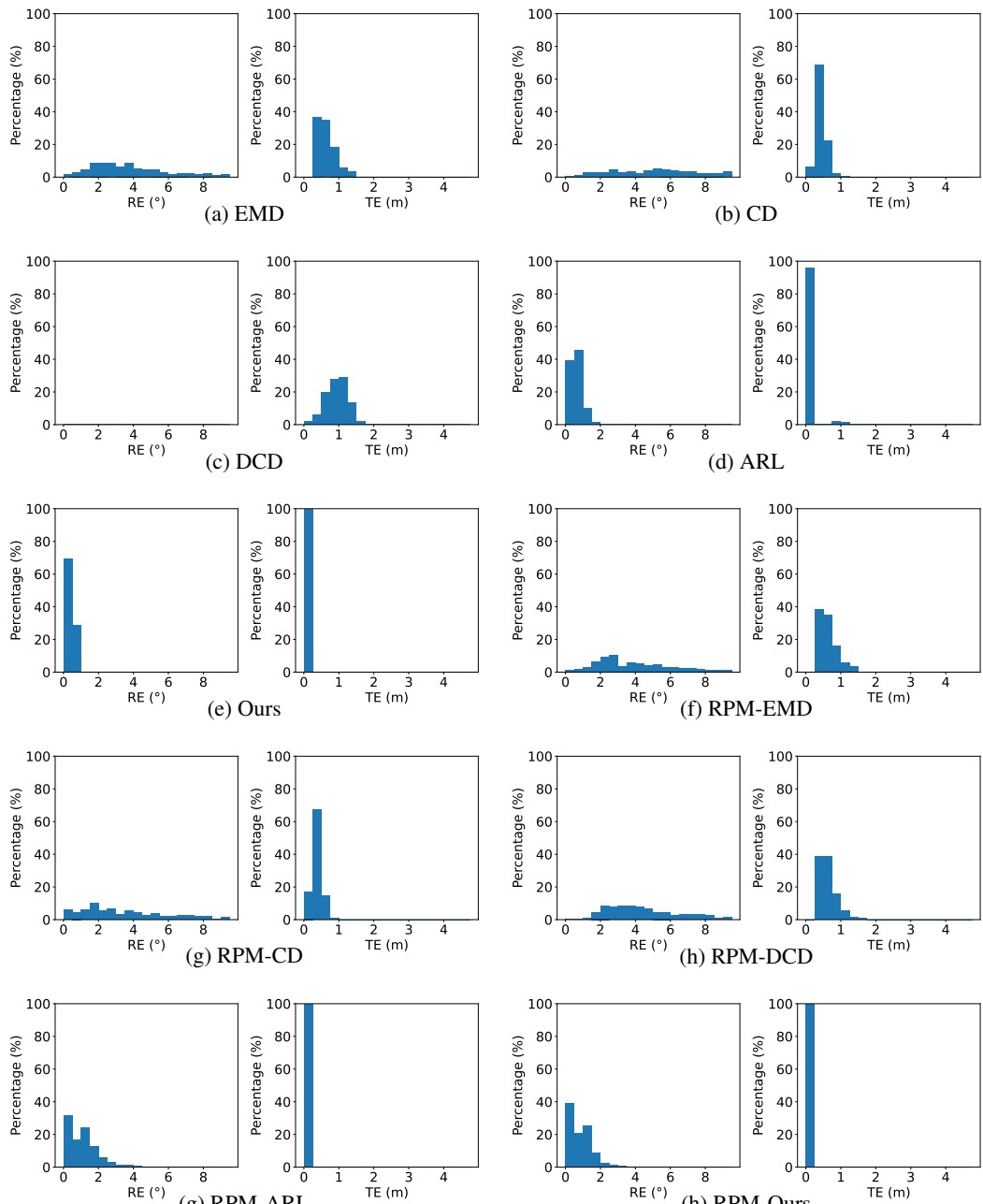

Figure 14: Histograms of RE and TE on the optimization-based and unsupervised learning-based registration methods. x-axis is RE($^\circ$) and TE(m), and y-axis is the percentage.

where $\mathcal{D}(\cdot, \cdot)$ is a typical point cloud distance metric, e.g., EMD, CD, or our DDF. After training, we adopt SVM to classify the global feature representations of point clouds to achieve classification.

**Visualization of the Decoded Point Clouds.** We also show the decoded results by the auto-decoder after trained with different distance metrics in Fig. 16, where it can be seen that the auto-encoder trained with our DDF can decode point clouds that are closer to input/ground-truth ones during inference, demonstrating its advantage.

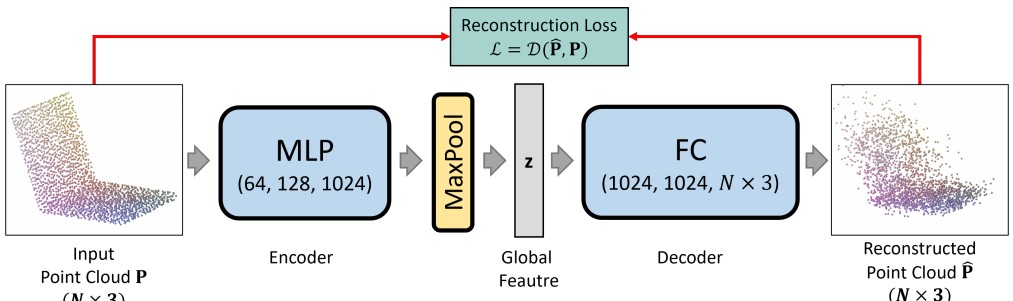

Figure 15: The network architecture of the auto-decoder for feature representation.

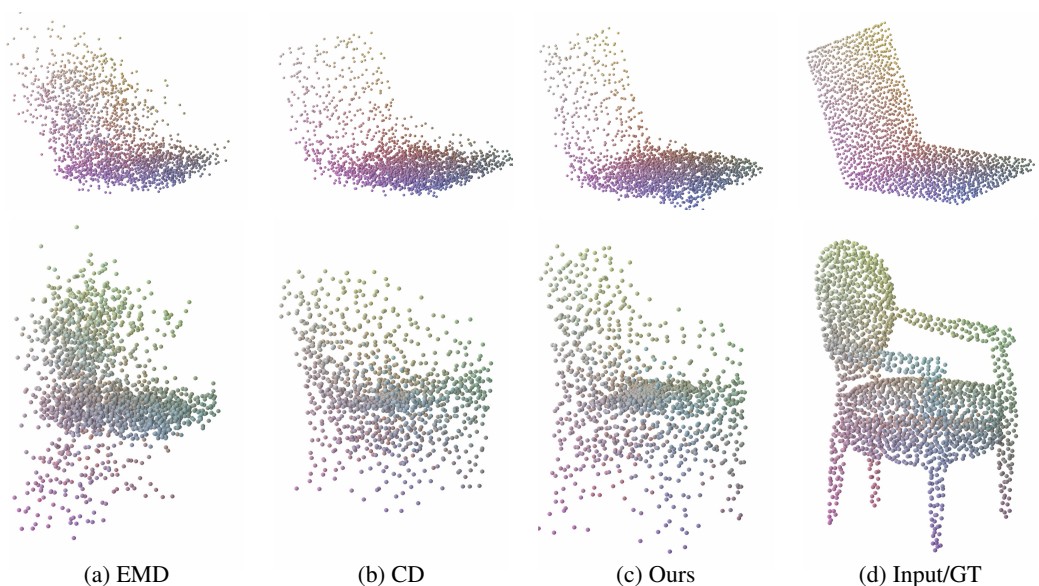

| (a) EMD | (b) CD | (c) Ours | (d) Input/GT |

Figure 16: Visual comparisons of decoded point clouds by the auto-encoder trained with different distance metrics.

