# OpenReview forum: "Directional Distance Field for Modeling the Difference between 3D Point Clouds"
_ICLR.cc/2024/Conference — Submitted to ICLR 2024_

### Official Review · Reviewer_pepa · 2023-10-30

**Soundness:** 3 good
**Presentation:** 2 fair
**Contribution:** 3 good
**Rating:** 6
**Confidence:** 5

**Summary:**

This paper introduces a new metric for measuring the distances between two point clouds. The idea is to compute the difference between the underlying 3D surfaces calibrated and induced by a set of reference points. The evaluations demonstrates the effectiveness of the proposed DDF loss.

**Strengths:**

1. The motivation makes sense. Previous losses (e.g. CD, EMD) focus on the point-to-point distances as the supervision, which brings lots of computational cost or easily reaches a local minimum. The proposed DDF loss measures the distance and directions to the underlying surface.

2. The performance seems good, which outperforms the widely used CD, EMD and DCD.

**Weaknesses:**

1. The name directional distance field is not suitable. I do not understand what is it until I finished reading the method section. The 'field' often indicates the signed distances or occupancies learned by a neural network. The proposed loss to measure distances from a reference point to the underline surface is not a 'field'.

2. The presentation can be improved.  I suggest that the authors to improve the writings in the introduction to make the readers understand the loss more easily. The inappropriate name 'directional distance field' and the unexplained 'reference point' make the introduction not clear enough. I can not understand the loss until I finish reading the method section, but finally I find the loss quite simple.

3. In Fig. 3, I find that DCD achieves quite good performances, why is the quantitative results of DCD in Tab. 1 that bad? More comparisons are needed.

4.  As shown in Fig.4, DDF is less efficient than CD and DCD. What is reason? Since CD measures point-to-point distances which should be much slower.

**Questions:**

It will be interest to see the performance of DDC under more downstream tasks like point cloud completion, point cloud generation, etc.

---

> ### Author Response · Authors · 2023-11-17
> **Responses to Reviewer pepa**
>
> We appreciate your acknowledgment of our efforts and constructive comments. In what follows, we address your comments comprehensively.
> ## **Comment 1.** *The name directional distance field is not suitable. I do not understand what is it until I finished reading the method section. The 'field' often indicates the signed distances or occupancies learned by a neural network. The proposed loss to measure distances from a reference point to the underline surface is not a 'field'.*
> **Response:** The field can also represent implicit fields calculated by classical methods, not limited to neural networks, such as IMLS [1], where Signed Distance Fields are estimated from point clouds using classical techniques without neural networks. Hence, utilizing 'Field' in our implicit representation of underlying surfaces is appropriate. \\
> The main idea of our distance metric is measuring the disparity between point clouds through the difference between the DDFs estimated from them, thus we denote our proposed DDF-based distance metric as 'DDF'.
> [1] Provably good moving least squares.
> ## **Comment 2.** *The presentation can be improved. I suggest that the authors to improve the writings in the introduction to make the readers understand the loss more easily. The inappropriate name 'directional distance field' and the unexplained 'reference point' make the introduction not clear enough. I can not understand the loss until I finish reading the method section, but finally I find the loss quite simple.*
> **Response:** At the start of Section 3, we include a figure in the revised manuscript to visually depict the overall concept of our distance metric, enhancing clarity and comprehension.
>
> ## **Comment 3.** *In Fig. 3, I find that DCD achieves quite good performances, why is the quantitative results of DCD in Tab. 1 that bad? More comparisons are needed.*
> **Response:** Table 1 of the manuscript shows the averaging numerical results of each category containing many 3D point clouds rather than only those displayed in the figure. In the updated manuscript, additional visual results have been included in Figure 11 within Appendix B. Furthermore, histograms depicting various evaluation metrics for different methods are presented in Figure 12, offering insights into the distribution of error and accuracy associated with each method.
>
> ## **Comment 4.** *As shown in Fig.4, DDF is less efficient than CD and DCD. What is reason? Since CD measures point-to-point distances which should be much slower.*
>
> **Response:** In our metric, we introduce a set of reference points whose quantity exceeds the number of points in the point clouds. This allows us to indirectly establish correspondences between two point clouds.Calculating the DDFs at the position of each reference point involves K-NN searching, which constitutes the most time-consuming aspect of our method. Tables 9 and 10 in our manuscript illustrate the relationship between the running time and the number of reference points, as well as the KNN size. Unlike CD and DCD, which establish correspondence for each point in the point cloud by finding the 1-NN point in the other point cloud, our method operates slightly slower. However, when considering both efficiency and effectiveness, our metric outperforms CD, DCD, and EMD.
>
> ## **Comment 5.** *It will be interest to see the performance of DDC under more downstream tasks like point cloud completion, point cloud generation, etc.*
>
> **Response:** Thank you for your valuable suggestion. Due to time constraints, we have conducted a preliminary exploration of the application of our distance metric in the point cloud completion task. We utilized the widely used framework PCN [2], in which we substituted the CD with our distance metric. We maintained the same experimental settings as the original PCN and selected several categories from the ShapeNet dataset for our experiments, namely Airplane, Car, Table, and Chair.
> The following table lists the numerical results, where it can be seen that the results from the network trained through our distance metric achieve higher F-Score than those from the network trained by CD.
>
>
> |Airplane| F-Score (0.01)|
> | ------ |  ------ |
> | CD | 0.8696 |
> | Ours | 0.8725 |
>
> |Car| F-Score (0.01)|
> | ------ | ------ |
> | CD   |0.7238 |
> | Ours  |0.7274 |
>
> |Sofa|F-Score (0.01)|
> | ------ | ------  |
> | CD | 0.5006 |
> | Ours | 0.5135 |
>
> |Table |F-Score (0.01)|
> | ------ | ------ |
> | CD | 0.6910 |
> | Ours | 0.7025 |
>
> [2] PCN: Point Completion Network.

---

### Official Review · Reviewer_UzJy · 2023-10-30

**Soundness:** 4 excellent
**Presentation:** 3 good
**Contribution:** 3 good
**Rating:** 8
**Confidence:** 3

**Summary:**

This paper proposes a new metric to measure the distance of two 3D point clouds.
The proposed method utilizes reference points to represent the local feature of the surface where point clouds should be. The distance is computed based on those reference points.
Experiments are performed on four downstream tasks, showing the proposed metric improves the performance of the methods of those downstream tasks.

**Strengths:**

1. It is reasonable to utilize the surface where the point clouds should be to compute the distance of two 3D point clouds.
2.The experiments show the proposed method improves the performance of all the downstream tasks.
3. The implementation of methods of all the downstream tasks are explained in detail.

**Weaknesses:**

The generation of reference points are not explained very clearly. The reviewer is confused by the shared identical weight operation and the reference point generation process.

**Questions:**

The reference points are generated from one of the two point clouds, and the weights in Equ.1 are computed for each kNN points and reference points. The reviewer wants to ask how to share the weights in g(q_m, P1) and g(q_m, P2)?

---

> ### Author Response · Authors · 2023-11-17
> **Responses to Reviewer UzJy**
>
> Thank you for acknowledging our work and offering constructive comments. We will respond comprehensively to your comments as follows.
> ## **Comment 1.** *The reviewer is confused by the shared identical weight operation. The reference points are generated from one of the two point clouds, and the weights in Equ.1 are computed for each kNN points and reference points. The reviewer wants to ask how to share the weights in g(q_m, P1) and g(q_m, P2)?*
> **Response:** As stated in Section 3.3 of the manuscript, the points within $\Omega(\mathbf{q},\mathbf{P}_1)$ and $\Omega(\mathbf{q},\mathbf{P}_2)$ are first sorted based on their distances to the reference point $\mathbf{q}$, i.e., $||\mathbf{p}{\tiny 1,1}-\mathbf{q}||\leq...\leq||\mathbf{p}{\tiny 1,K}-\mathbf{q}||$ with $\mathbf{p}{\tiny 1,1},...,\mathbf{p}{\tiny 1,K}\in\Omega(\mathbf{q},\mathbf{P}_1)$ and $||\mathbf{p}{\tiny 2,1}-\mathbf{q}||\leq...\leq||\mathbf{p}{\tiny 2,K}-\mathbf{q}||$ with $\mathbf{p}{\tiny 2,1},...,\mathbf{p}{\tiny 2,K}\in\Omega(\mathbf{q},\mathbf{P}_2)$. This sorting process facilitates the transfer of weights from the target/ground-truth point cloud to the source/generated cloud. Specifically, for the calculation of directional distances across both point clouds, we adhere to the relation $w(\mathbf{q},\mathbf{p}{\tiny 2,k})=w(\mathbf{q},\mathbf{p}{\tiny 1,k}),\ k=1,...,K$. The weights can be computed from $\mathbf{P}_1$ (resp. $\mathbf{P}_2$) and then shared with $\mathbf{P}_2$ (resp. $\mathbf{P}_1$), according to the task.
> Our weight-sharing strategy is built upon the observation that if two point clouds are exactly the same, the weights obtained through the same calculation method should be equal for a typical reference point. If the weights are derived independently using the inverse distance for two point clouds, for the point cloud to be optimized/generated, the directional distance will be a high-order non-linear function of its points, thereby complicating the optimization process. The ablative study results in Table 7 of our manuscript show the superiority of the shared weights.
>
> ## **Comment 2.** *The generation of reference points are not explained very clearly.*
> **Response:** As explained in Section 3.2, the reference points serve to assess the disparity between the DDFs of the point clouds and are distributed in proximity to the implicit surfaces. Since each point in the point cloud is located on its underlying surface, we introduce offsets of Gaussian noise to displace these points away from the underlying surface while keeping them in close proximity. The standard deviations for the Gaussian noise are adjusted based on their distances to the nearest points in the point cloud, considering the non-uniformity of the point cloud. In the updated pdf file (i.e., the revised manuscript), we added one more figure, i.e., Fig. 3, to visually illustrate the generation process to help understanding.

---

### Official Review · Reviewer_4UtW · 2023-10-30

**Soundness:** 2 fair
**Presentation:** 2 fair
**Contribution:** 1 poor
**Rating:** 5
**Confidence:** 4

**Summary:**

This work presents a point cloud distance function as an alternative to EMD and CD. The proposed distance is theoretically superior to EMD and CD as it better describes underlying surfaces. The experimental result over several tasks (3D shape reconstruction, rigid registration, scene flow estimation and feature representation) demonstrates this.

**Strengths:**

I believe the idea of using better surface descriptions for point cloud related tasks is relevant and this paper show improvements in several related tasks including shape reconstruction, rigid registration, scene flow estimation, and feature representation.

Despite some comments, the distance function is technically sound.

Paper is generally well presented.

**Weaknesses:**

This work should add comparisons against relevant shape or surface descriptors in the literature. For example, against 3D shape context (and other methods) in Frome et al. ("Recognizing objects in range data using regional point descriptors." ECCV 2004). This lack of comparison is an important weakness as the proposed method resembles the common pipeline of
1)  Computing keypoints, here by using a sampling mechanism plus noise;
2) Obtaining descriptors at each keypoint, here as the concatenation of the magnitude and direction of a sum of weighted distances between a keypoint and the K-NN of a point cloud.

The proposed distance aggregates descriptor distances. Here, descriptor correspondences result from sharing the same keypoints to obtain each set of descriptors.

Another weakness is in formulations that should be presented more clearly, perhaps improving notation. For example, it is unclear why "g(qm, P1) and g(qm, P2) share identical weights". From equations, they seem to be different as calculated over different point clouds.

Additional

In Sec. 4.2,  R should be in SO(3).

Also, there is some abuse of notation when applying a rotation over a point cloud.

The registration methods Opt-EMD, Opt-CD, Opt-ARL and Opt-Ours require better explanation.

**Questions:**

Why do you propose g = [f, v] instead of a 3-vector f*v?

---

> ### Author Response · Authors · 2023-11-17
> **Responses to Reviewer 4UtW (1/2)**
>
> We appreciate your acknowledgment of our work and the constructive feedback you've provided. In the subsequent discussion, we will thoroughly address the concerns you have raised.
> ## **Comments 1.** *This work should add comparisons against relevant shape or surface descriptors in the literature.*
> **Response:** The reviewer may have **misunderstood** our Directional Distance Field (DDF) to some extent, resulting in the comparison with the mentioned surface descriptors not making sense. The detailed reasons are listed as follows.
> * The reviewer mentions the 3D Shape Context as a surface descriptor, which employs histograms to characterize local point cloud structures. However, this descriptor is **non-differentiable**, making it unsuitable for replacing our differentiable distance function  in the distance metric. Additionally, other differentiable descriptors, including learning-based ones, are **not viable** alternatives for the distance metric. This limitation essentially arises from these descriptors explicitly characterizing local surface structures, defined on surfaces where point clouds are sampled. In contrast, our DDF serves as an implicit field, representing underlying surfaces as its iso-surface with a specific value. Consequently, the DDF is defined across the entire 3D space, allowing it to be used at any point.
> * The key points used in the surface descriptor **differ** from the reference points used in our method. Key points, usually a subset of the whole point cloud, are selected to reduce computational complexity while retaining crucial information. And the correspondence between key points in different clouds is established through feature matching. Differently, the reference points in our method are located near the underlying surfaces and they are used to compute the difference between the DDFs of the point clouds, where the correspondence is established naturally according to these shared reference points.
>
> ## **Comment 2.** *Another weakness is in formulations that should be presented more clearly, perhaps improving notation. For example, it is unclear why "g(qm, P1) and g(qm, P2) share identical weights". From equations, they seem to be different as calculated over different point clouds.*
> **Response:** As stated in Sec. 3.3, the points within $\Omega(\mathbf{q},\mathbf{P}_1)$ and $\Omega(\mathbf{q},\mathbf{P}_2)$ are first sorted based on their distances to the reference point $\mathbf{q}$, i.e., $||\mathbf{p}{\tiny{1,1}}-\mathbf{q}||\leq...\leq||\mathbf{p}{\tiny{1,K}}-\mathbf{q}||$ with $\mathbf{p}{\tiny{1,1}},...,\mathbf{p}{\tiny{1,K}}\in\Omega(\mathbf{q},\mathbf{P}_1)$ and $||\mathbf{p}{\tiny{2,1}}-\mathbf{q}||\leq...\leq||\mathbf{p}{\tiny{2,K}}-\mathbf{q}||$ with $\mathbf{p}{\tiny{2,1}},...,\mathbf{p}{\tiny{2,K}}\in\Omega(\mathbf{q},\mathbf{P}_2)$. This sorting process facilitates the transfer of weights from the target/ground-truth point cloud to the source/generated cloud. Specifically, for the calculation of directional distances across both point clouds, we adhere to the relation $w(\mathbf{q},\mathbf{p}{\tiny{2,k}})=w(\mathbf{q},\mathbf{p}{\tiny{1,k}}),\ k=1,...,K$. The weights can be computed from $\mathbf{P}_1$ (resp. $\mathbf{P}_2$) and then shared with $\mathbf{P}_2$ (resp. $\mathbf{P}_1$), according to the task.
> Our weight-sharing strategy is built upon the observation that if two point clouds are exactly the same, the weights obtained through the same calculation method should be equal for a typical reference point. If the weights are derived independently using the inverse distance for two point clouds, for the point cloud to be optimized/generated, the directional distance will be a high-order non-linear function of its points, thereby complicating the optimization process. The experimental results in Table 7 of our manuscript show the superiority of the shared weights.
> Eqs. (1) and (2) represent the calculation of our designed DDF. In Section 3.4 of the updated pdf file, we clarified the motivation of the shared weights.
>
> ## **Comment 3.** *In Sec. 4.2, R should be in SO(3). Also, there is some abuse of notation when applying a rotation over a point cloud.*
> **Response:**  In Section 4.2, we stated that '$\mathbf{R}\in\mathbb{R}^{3\times 3}$ is the rotation matrix', and this is equivalent to '$\mathbf{R}\in\texttt{SO}(3)$'. As the reviewer rightly noted, the latter representation is more concise. We updated the manuscript with this form.
> Thanks to the reviewer for the reminder. We have identified an inaccurate expression in the matrix multiplication $\mathbf{R}\mathbf{P}{\tiny src}$ in Eq. (5) of the manuscript, where $\mathbf{R}\in\mathbb{R}^{3\times 3}$ and $\mathbf{P}{\tiny src}\in\mathbb{R}^{N_{\small src}\times 3}$. To rectify this, we introduce a new symbol to denote the rigid transformation on the point cloud in Eq. (5) of the revised manuscript, denoted as $\mathcal{T}(\mathbf{P}{\tiny src},\mathbf{R},\mathbf{t})$.

---

> > ### Author Response · Authors · 2023-11-17
> > **Responses to Reviewer 4UtW (2/2)**
> >
> > ## **Comment 4.** *The registration methods Opt-EMD, Opt-CD, Opt-ARL and Opt-Ours require better explanation.*
> > **Response:** In the figure, 'Opt-X' is an abbreviation for the Optimized Method based on Eq. (5), where a typical distance metric is used, i.e., 'X' can be CD, EMD, ARL, or our DDF. We have clarified this issue in the updated pdf file.
> > ## **Comment 5.** *Why do you propose g = [f, v] instead of a 3-vector fv?*
> > ** Response:** We are concerned that the reviewer **may have misunderstood** the computation of $\mathbf{v}$ in Eq. (2) of our manuscript. The vector $\mathbf{v}$ is the weighted sum of vectors pointing from  the reference point $\mathbf{q}$ to each $K$-NN point $\mathbf{p}_k$, and it is **not normalized**. Therefore, the '$\mathbf{v}$' mentioned in our manuscript is indeed 'v*f' as pointed out by the reviewer. Utilizing only '$\mathbf{f}$' or '$\mathbf{v}$' alone would result in **decreased performance**, as demonstrated in Table 8 of our manuscript. As the term 'direction' is typically associated with normalization, which might lead to confusion, we replace 'direction from each ...' with 'vector pointing from each ...' in the final version of the manuscript.

---

### Author Response · Authors · 2023-11-17

We extend our gratitude to the reviewers for dedicating their valuable time and effort to assess our work and for acknowledging its merit. In response to the reviewers' comments, we have diligently addressed their feedback and updated our manuscript accordingly (i.e., the updated pdf file).
The key enhancements include:
* In Section 3.1, we have incorporated a new figure that provides a **comprehensive overview** of the proposed distance metric, enhancing the overall illustration.
* Within Section 3.2, a new figure has been included to elucidate the **process of generating reference points**, offering greater clarity on this aspect of our work.
* Section 3.4 now contains additional **details and motivation** pertaining to the shared weights, enriching the understanding of this particular aspect of our methodology.
* In Section 4.2, we have rectified the **mathematical formulation** of the rigid transformation, ensuring accuracy and precision in our presentation.
* Figure 7 has been augmented with an **explanation of "Opt-X"**, providing further insights into this element of our research.
* Within Appendix Section B, we have **expanded** the results of the shape reconstruction experiment by incorporating **additional visual results** and presenting a more comprehensive **error distribution analysis**.

We believe that these updates significantly enhance the quality and clarity of our manuscript, addressing the valuable comments provided by the reviewers. We appreciate the opportunity to refine our work based on their insightful comments and we believe our detailed responses in the following clearly address the reviewers' concerns.

---

> ### Author Response · Authors · 2023-11-21
> **Looking Forward to Your Feedback. Thanks.**
>
> Dear Reviewers
>
> Thank you for dedicating your time and effort to reviewing our work and providing constructive comments and favorable recommendations. We have carefully considered and addressed all the concerns you raised in your review, as outlined in our response and reflected in the updated manuscript. As the Reviewer-Author discussion phase is nearing its conclusion, we eagerly await any further feedback from you. Should you have any additional questions, we would be delighted to provide detailed responses.

---

> > ### Author Response · Authors · 2023-11-23
> >
> > Dear Reviewers,
> >         Thank you for dedicating your time and effort to reviewing our work and reading our responses. As the Reviewer-Author discussion phase is nearing its conclusion, we eagerly await any further feedback from you.
> >
> > Best regards,
> > The authors

---

### Meta-Review · Area_Chair_9JZi · 2023-12-07

**Metareview:**

A distance metric called directional distance field (DDF) is proposed. By associating each reference point with two given point clouds through computing its directional distances to them, the difference in directional distances of an identical reference point characterizes the geometric difference between a typical local region of the two point clouds. Finally, DDF is obtained by averaging the directional distance differences of all reference points.

The paper received one “marginally below the acceptance threshold” rating, one “accept” rating, and one “marginally above the acceptance threshold” rating.

Reviewer 4UtW thinks this work should add comparisons against relevant shape or surface descriptors and another weakness is in formulations that should be presented more clearly. But the authors think the comparisons are unnecessary to be added. The reasons given are 1) 3D shape context as a surface descriptor is 3D Shape Context as a surface descriptor; 2) The key points used in the surface descriptor differ from the reference points. Comparisons are not necessary to make by using the same flowchart. The aim of the paper is to quantify the dissimilarity between two unstructured 3D point clouds. The reviewer thinks 3D shape context can make such a task. Those reasons not to make comparisons are unconvincing.

Reviewer UzJy thinks the generation of reference points are not explained very clearly. Reviewer UzJy is confused by the shared identical weight operation and the reference point generation process. The authors gave detailed responses to the comments.

Reviewer pepa thinks the name directional distance field is not suitable, the presentation can be improved, more comparisons are needed, and DDF is less efficient. The authors gave responses to the comments. The authors include a figure for clarity but the writing is still not clear. The figure is not well understood for a clear presentation. I have also a puzzle on the reference points. How to take the reference points? What a coordinate system are they under? Is the system local or global? If two point clouds P1 and P2 are of different scales or densities but from a same shape object, is the method effective? The proposed distance metric is simple and its usage is limited. The current metric may be only suitable to two similar point clouds of object levels. But this is not a difficult task. The further studies and revisions are still needed.

Based on the above comments, the decision was to reject the paper.

**Justification For Why Not Higher Score:**

Reviewer 4UtW thinks this work should add comparisons against relevant shape or surface descriptors and another weakness is in formulations that should be presented more clearly. But the authors think the comparisons are unnecessary to be added. The reasons given are 1) 3D shape context as a surface descriptor is 3D Shape Context as a surface descriptor; 2) The key points used in the surface descriptor differ from the reference points. Comparisons are not necessary to make by using the same flowchart. The aim of the paper is to quantify the dissimilarity between two unstructured 3D point clouds. The reviewer thinks 3D shape context can make such a task. Those reasons not to make comparisons are unconvincing.

Reviewer UzJy thinks the generation of reference points are not explained very clearly. Reviewer UzJy is confused by the shared identical weight operation and the reference point generation process. The authors gave detailed responses to the comments.

Reviewer pepa thinks the name directional distance field is not suitable, the presentation can be improved, more comparisons are needed, and DDF is less efficient. The authors gave responses to the comments. The authors include a figure for clarity but the writing is still not clear. The figure is not well understood for a clear presentation. I have also a puzzle on the reference points. How to take the reference points? What a coordinate system are they under? Is the system local or global? If two point clouds P1 and P2 are of different scales or densities but from a same shape object, is the method effective? The proposed distance metric is simple and its usage is limited. The current metric may be only suitable to two similar point clouds of object levels. But this is not a difficult task. The further studies and revisions are still needed.

**Justification For Why Not Lower Score:**

N/A

---

### Decision · Program_Chairs · 2024-01-16

Reject